# DROPOUT: EXPLICIT FORMS AND CAPACITY CONTROL

## ABSTRACT

We investigate the capacity control provided by dropout in various machine learning problems. First, we study dropout for matrix sensing, where it induces a data-dependent regularizer that, in expectation, equals the weighted trace-norm of the product of the factors. In deep learning, we show that the data-dependent regularizer due to dropout directly controls the Rademacher complexity of the underlying class of deep neural networks. These developments enable us to give concrete generalization error bounds for the dropout algorithm in both matrix completion as well as training deep neural networks. We evaluate our theoretical findings on real-world datasets, including MovieLens, Fashion MNIST, and CIFAR-10.

## 1 INTRODUCTION

Dropout is a popular regularization technique for training deep neural networks that aims at "breaking co-adaptation" among neurons by randomly dropping them at training time (Hinton et al., 2012). Dropout has been shown effective across a wide range of machine learning tasks, from classification (Srivastava et al., 2014; Szegedy et al., 2015) to regression (Toshev & Szegedy, 2014). Notably, it is considered as a major component in the design of AlexNet (Krizhevsky et al., 2012), which won the prominent ImageNet challenge in 2012 with a significant margin and helped transform the field of computer vision.

Following the empirical success of dropout, there have been several studies in recent years that focus on understanding theoretical underpinnings of dropout (Baldi & Sadowski, 2013; Wager et al., 2013; McAllester, 2013; Van Erven et al., 2014; Helmbold & Long, 2015; Gal & Ghahramani, 2016; Gao & Zhou, 2016; Mou et al., 2018; Bank & Giryes, 2018; Cavazza et al., 2018; Mianjy et al., 2018). However, none of these works adequately address the following basic question: *how does dropout control the capacity of deep neural networks?* In this paper, we provide an answer to this question. We focus on the task of *deep regression* using squared $\ell_2$-loss which yields state-of-the-art results in human pose estimation (Toshev & Szegedy, 2014), facial landmark detection, age estimation (Lathuilière et al., 2019), and more. We give precise generalization bounds for deep regression with dropout; our bounds leverage recent results in understanding generalization in deep learning and help explain practice with meaningful bounds. Along the way we also recover the state-of-the-art bounds for matrix completion. The key contributions in this paper are as follows.

1. We introduce dropout for matrix completion, a procedure that randomly drops the columns of the factors during training. Interestingly, this algorithmic procedure induces a data-dependent regularizer that, in expectation, equals weighted trace-norm – a complexity measure that enjoys strong generalization guarantees (Foygel et al., 2011).

2. For two-layer neural networks with ReLU activation, when the input distribution is symmetric and isotropic, we show that dropout induces the $\ell_2$ path-norm which has been shown to provide scale-sensitive generalization guarantees in deep neural networks (Neyshabur et al., 2015).

3. We study deep regression, where a feed-forward neural network is trained with dropout under squared loss. With no further assumptions on the data distribution and the network architecture, we show that the dropout regularizer is a data-dependent measure whose expected value serves as a strong complexity measure for networks trained with dropout. In

particular, we show that a network with a small dropout regularizer enjoys a small generalization gap.

4. We empirically evaluate our theoretical findings for matrix completion and deep regression on real world datasets MovieLens, Fashion MNIST and CIFAR-10.

## 1.1 NOTATION

We denote matrices, vectors, scalar variables and sets by Roman capital letters, Roman small letters, small letters, and script letters, respectively (e.g. X, x, $x$, and $\mathcal{X}$). For any integer $d$, we represent the set $\{1, \ldots, d\}$ by $[d]$. For any vector $x \in \mathbb{R}^d$, $\text{diag}(x) \in \mathbb{R}^{d \times d}$ represents the diagonal matrix with diagonal elements equal to x, and $\sqrt{x}$ is the elementwise squared root of x. Let $\|x\|$ represent the $\ell_2$-norm of vector x, and $\|X\|$, $\|X\|_F$, and $\|X\|_*$ represent the spectral norm, the Frobenius norm, and the nuclear norm of matrix X, respectively. Similarly, given a positive definite matrix C, we denote the Mahalonobis norm as $\|x\|_C^2 = x^\top C x$. The standard inner product is represented by $\langle \cdot, \cdot \rangle$, for vectors or matrices, where $\langle X, X' \rangle = \text{Tr}(X^\top X')$.

## 2 MATRIX SENSING

We begin with understanding dropout for matrix sensing, a problem which arguably is an important instance of a matrix learning problem with lots of applications, and is well understood from a theoretical perspective. Here is the problem setup.

Let $M_* \in \mathbb{R}^{d_2 \times d_0}$ be a matrix with rank $r_* := \text{Rank}(M_*)$. Let $A^{(1)}, \ldots, A^{(n)}$ be a set of measurement matrices of the same size as M. The goal of matrix sensing is to recover the matrix $M_*$ from $n$ observations of the form $y_i = \langle M_*, A^{(i)} \rangle$ such that $n \ll d_2 d_0$. A natural approach to solve this problem is to enforce the rank constraints implicitly using the Burer-Monteiro factorization and to solve the following empirical risk minimization problem:

$$\min_{U \in \mathbb{R}^{d_2 \times d_1}, V \in \mathbb{R}^{d_0 \times d_1}} \widehat{L}(U, V) := \frac{1}{n} \sum_{i=1}^{n} (y_i - \langle UV^\top, A^{(i)} \rangle)^2. \tag{1}$$

When $d_1 \gg r_*$, there exist many "bad" empirical minimizers, i.e., those with a large true risk. However, recently, Li et al. (2018) showed that under restricted isometry property, despite the existence of such poor ERM solutions, gradient descent with proper initialization is *implicitly* biased towards finding solutions with minimum nuclear norm – this is an important result which was first conjectured and empirically verified by Gunasekar et al. (2017). However, for the most part, modern machine learning systems employ *explicit* regularization techniques. In fact, as we show in the experimental section, the *implicit* bias due to (stochastic) gradient descent does not prevent it from blatant overfitting in the matrix completion problem. We argue that in such cases, there is a need for "suitable" *explicit* regularization, such as trace-norm Recht et al. (2010); Candès & Recht (2009), weighted trace-norm Foygel et al. (2011), or max norm Srebro & Shraibman (2005).

We propose solving the ERM problem (1) with algorithmic regularization due to dropout, where at training time, columns of U and V are dropped uniformly at random. As opposed to the *implicit* effect of gradient descent, this dropout heuristic *explicitly* regularizes the empirical objective. It is then natural to ask, in the case of matrix sensing, if dropout also biases the ERM towards certain low norm solutions. To answer this question, we begin with the observation that dropout can be viewed as an instance of SGD on the following objective:

$$\widehat{L}_{\text{drop}}(U, V) = \frac{1}{n} \sum_{j=1}^{n} \mathbb{E}_B (y_j - \langle UBV^\top, A^{(j)} \rangle)^2, \tag{2}$$

where $B \in \mathbb{R}^{d_1 \times d_1}$ is a diagonal matrix whose diagonal elements are Bernoulli random variables distributed as $B_{ii} \sim \frac{1}{1-p} \text{Ber}(1-p)$. In this case, it is easy to show that for any $p \in [0, 1)$:

$$\widehat{L}_{\text{drop}}(U, V) = \widehat{L}(U, V) + \frac{p}{1-p} \widehat{R}(U, V), \quad \widehat{R}(U, V) = \sum_{i=1}^{d_1} \frac{1}{n} \sum_{j=1}^{n} (u_i^\top A^{(j)} v_i)^2$$

where $\widehat{R}(\mathrm{U}, \mathrm{V})$ is a data-dependent term that captures the *explicit* regularizer due to dropout (see Proposition 1 in the Appendix). Provided that the sample size $n$ is large enough, the *explicit* regularizer is well concentrated around its mean (see Lemma 2 in the Appendix). Further, given that we seek a minima of $\widehat{L}_{\mathrm{drop}}$, it suffices to consider the factors with the minimal value of the regularizer among all that yield the same empirical loss. This motivates studying the the following distribution-dependent *induced* regularizer:

$$\Theta(\mathrm{M}) := \min_{\mathrm{UV}^\top = \mathrm{M}} R(\mathrm{U}, \mathrm{V}), \quad \text{where} \quad R(\mathrm{U}, \mathrm{V}) := \mathbb{E}_\mathrm{A}[\widehat{R}(\mathrm{U}, \mathrm{V})].$$

Surprisingly, for a wide range of random measurements, $\Theta(\cdot)$ turns out to be a "suitable" regularizer. Here, we instantiate two important examples (see Proposition 2 in the Appendix).

**Gaussian measurements.** For all $j \in [n]$, let $\mathrm{A}^{(j)}$ be standard Gaussian matrices. Then $\Theta(\mathrm{M}) = \frac{1}{d_1} \|\mathrm{M}\|_*^2$ is the standard trace-norm regularization studied in Srebro et al. (2005); Bach (2008); Candès & Tao (2009).

**Matrix completion.** For all $j \in [n]$, let $\mathrm{A}^{(j)}$ be an indicator matrix whose $(i, k)$-th element is selected randomly with probability $p(i)q(k)$, where $p(i)$ and $q(k)$ denote the probability of choosing the $i$-th row and the $j$-th column, respectively. Then

$$\Theta(\mathrm{M}) = \frac{1}{d_1} \| \operatorname{diag}(\sqrt{p}) \mathrm{UV}^\top \operatorname{diag}(\sqrt{q}) \|_*^2$$

is the *weighted trace-norm* studied by Srebro & Salakhutdinov (2010) and Foygel et al. (2011).

These observations are specifically important because they connect dropout, an algorithmic heuristic in deep learning, to strong complexity measures that are empirically effective as well as theoretically well understood. To illustrate, here we give a generalization bound for matrix completion with dropout in terms of the value of the *explicit* regularizer at the minimum of the empirical problem.

**Theorem 1.** Without loss of generality, assume that $d_2 \geq d_0$ and $\|\mathrm{M}_*\| \leq 1$. Furthermore, assume that $\min_{i,j} p(i)q(j) \geq \frac{\log(d_2)}{n\sqrt{d_2 d_0}}$. Let $(\mathrm{U}, \mathrm{V})$ be a minimizer of the dropout ERM objective in equation (2), and assume that $\max_i \|\mathrm{U}(i,:)\|^2 \leq \gamma$, $\max_i \|\mathrm{V}(i,:)\|^2 \leq \gamma$. Let $\alpha$ be such that $\widehat{R}(\mathrm{U}, \mathrm{V}) \leq \alpha/d_1$. Then, for any $\delta \in (0, 1)$, the following generalization bounds holds with probability at least $1 - 2\delta$ over a sample of size $n$:

$$L(\mathrm{U}, \mathrm{V}) \leq \widehat{L}(\mathrm{U}, \mathrm{V}) + C(1 + \gamma)\sqrt{\frac{\alpha d_2 \log(d_2)}{n}} + C'(1 + \gamma^2)\sqrt{\frac{\log(2/\delta)}{2n}}$$

as long as $n = \Omega\left((d_1 \gamma^2/\alpha)^2 \log(2/\delta)\right)$, where $C, C'$ are some absolute constants.

The proof of Theorem 1 follows from standard generalization bounds for $\ell_2$ loss (Mohri et al., 2018) based on the Rademacher complexity (Bartlett & Mendelson, 2002) of the class of functions with weighted trace-norm bounded by $\sqrt{\alpha}$, i.e. $\mathcal{M}_\alpha := \{\mathrm{M} : \| \operatorname{diag}(\sqrt{\mathrm{p}})\mathrm{M} \operatorname{diag}(\sqrt{\mathrm{q}})\|_*^2 \leq \alpha\}$. A bound on the Rademacher complexity of this class was established by Foygel et al. (2011). The technicalities here include showing that the explicit regularizer is well concentrated around its expected value, as well as deriving a bound on the supremum of the predictions. A few remarks are in order.

We require that the sampling distributions be non-degenerate, as specified by the condition $\min_{i,j} p(i)q(j) \geq \frac{\log(d_2)}{n\sqrt{d_2 d_0}}$. This is a natural requirement for bounding the Rademacher complexity of $\mathcal{M}_\alpha$, as discussed in Foygel et al. (2011).

We note that for large enough sample size, $\widehat{R}(\mathrm{U}, \mathrm{V}) \approx R(\mathrm{U}, \mathrm{V}) \approx \Theta(\mathrm{UV}^\top) = \frac{1}{d_1} \| \operatorname{diag}(\sqrt{p})\mathrm{UV}^\top \operatorname{diag}(\sqrt{q})\|_*^2$, where the second approximation is due the fact that the pair $(\mathrm{U}, \mathrm{V})$ is a minimizer. That is, compared to the weighted trace-norm, the value of the explicit regularizer at the minimizer roughly scales as $1/d_1$. Hence the assumption $\widehat{R}(\mathrm{U}, \mathrm{V}) \leq \alpha/d_1$ in the statement of the corollary.

In practice, for models that are trained with dropout, the training error $\widehat{L}(\mathrm{U}, \mathrm{V})$ is negligible (see Figure 1 for experiments on the MovieLens dataset). Moreover, given that the sample size is large

enough, the third term can be made arbitrarily small. Having said that, the second term, which is $\tilde{O}(\gamma\sqrt{\alpha d_2/n})$, dominates the right hand side of generalization error bound in Theorem 6.

The assumption $\max_i \|U(i,:)\|^2 \leq \gamma$, $\max_i \|V(i,:)\|^2 \leq \gamma$ is motivated by the practice of deep learning; such *max-norm* constraints are typically used with dropout, where the norm of the vector of incoming weights at each hidden unit is constrained to be bound by a constant (Srivastava et al., 2014). In this case, if a dropout update violates this constraint, the weights of the hidden unit are projected back to the constraint norm ball. In proofs, we need this assumption to give a concentration bound for the empirical explicit regularizer, as well as bound the supremum deviation between the predictions and the true values. We remark that the value of $\gamma$ also determines the complexity of the function class. On one hand, the generalization gap explicitly depends on and increases with $\gamma$. However, when $\gamma$ is large, the constraints on $U, V$ are milder, so that $\widehat{L}(U,V)$ can be made smaller.

Finally, the required sample size heavily depends on the value of the explicit regularizer at the optima $(\alpha/d_1)$, and hence, on the dropout rate $p$. In particular, increasing the dropout rate increases the regularization parameter $\lambda := \frac{p}{1-p}$, thereby intensifies the penalty due to the explicit regularizer. Intuitively, a larger dropout rate $p$ results in a smaller $\alpha$, thereby a tighter generalization gap can be guaranteed. We show through experiments that that is indeed the case in practice.

## 3 DEEP NEURAL NETWORKS

Next, we focus on neural networks with multiple hidden layers. Let $\mathcal{X} \subseteq \mathbb{R}^{d_0}$ and $\mathcal{Y} \subseteq \mathbb{R}^{d_k}$ denote the input and output spaces, respectively. Let $\mathcal{D}$ denote the joint probability distribution on $\mathcal{X} \times \mathcal{Y}$. Given $n$ examples $\{(x_i, y_i)\}_{i=1}^n \sim \mathcal{D}^n$ drawn i.i.d. from the joint distribution and a loss function $\ell : \mathcal{Y} \times \mathcal{Y} \to \mathbb{R}$, the goal of learning is to find a hypothesis $f_w : \mathcal{X} \to \mathcal{Y}$, parameterized by w, that has a small *population risk* $L(w) := \mathbb{E}_{\mathcal{D}}[\ell(f_w(x), y)]$.

We focus on the squared $\ell_2$ loss, i.e., $\ell(y, y') = \|y - y'\|^2$, and study the generalization properties of the dropout algorithm for minimizing the *empirical risk* $\widehat{L}(w) := \frac{1}{n}\sum_{i=1}^n [\|y_i - f_w(x_i)\|^2]$. We consider the hypothesis class associated with feed-forward neural networks with $k$ layers, i.e., functions of the form $f_w(x) = W_k\sigma(W_{k-1}\sigma(\cdots W_2\sigma(W_1x)\cdots))$, where $W_i \in \mathbb{R}^{d_i \times d_{i-1}}$, for $i \in [k]$, is the weight matrix at $i$-th layer. The parameter w is the collection of weight matrices $\{W_k, W_{k-1}, \ldots, W_1\}$ and $\sigma : \mathbb{R} \to \mathbb{R}$ is an activation function applied entrywise to an input vector.

In modern machine learning systems, rather than talk about a certain network topology, we should think in terms of layer topology where each layer could have different characteristics – for example, fully connected, locally connected, or convolutional. In convolutional neural networks, it is a common practice to apply dropout only to the fully connected layers and not to the convolutional layers. Furthermore, in deep regression, it has been observed that applying dropout to only one of the hidden layers is most effective (Lathuilière et al., 2019). In our study, dropout is applied on top of the learned representations or *features*, i.e. the output of the top hidden layer. In this case, dropout updates can be viewed as stochastic gradient descent iterates on the *dropout objective*:

$$\widehat{L}_{\text{drop}}(w) := \frac{1}{n}\sum_{i=1}^n \mathbb{E}_B\|y_i - W_kB\sigma(W_{k-1}\sigma(\cdots W_2\sigma(W_1x_i)\cdots))\|^2 \qquad \text{(dropout objective)}$$

where B is a diagonal random matrix with diagonal elements distributed identically and independently as $B_{ii} \sim \frac{1}{1-p}\text{Bern}(1-p)$, $i \in [d_{k-1}]$, for some *dropout rate* $p$. We seek to understand the *explicit* regularizer due to dropout:

$$\widehat{R}(w) := \widehat{L}_{\text{drop}}(w) - \widehat{L}(w) \qquad \text{(explicit regularizer)}$$

We denote the output of the $i$-th hidden node in the $j$-th hidden layer on an input vector x by $a_{i,j}(x) \in \mathbb{R}$; for example, $a_{1,2}(x) = \sigma(W_2(1,:)^\top\sigma(W_1x))$. Similarly, the vector $a_j(x) \in \mathbb{R}^{d_j}$ denotes the activation of the $j$-th layer on input x. Using this notation, we can conveniently rewrite the dropout objective as $\widehat{L}_{\text{drop}}(w) := \frac{1}{n}\sum_{i=1}^n \mathbb{E}_B\|y_i - W_kBa_{k-1}(x_i)\|^2$. It is then easy to show that the explicit regularizer due to dropout is given as (see Propostion 3 in the Appendix):

$$\widehat{R}(w) = \frac{p}{1-p}\sum_{j=1}^{d_{k-1}} \|W_k(:,j)\|^2\widehat{a}_j^2, \quad \text{where } \widehat{a}_j = \sqrt{\frac{1}{n}\sum_{i=1}^n a_{j,k-1}(x_i)^2}.$$

The explicit regularizer $\widehat{R}(\mathrm{w})$ is the summation over hidden nodes, of the product of the squared norm of the outgoing weights with the empirical second moment of the output of the corresponding neuron. For a two layer neural network with ReLU, when the input distribution is symmetric and isotropic, the expected regularizer is equal to (see Proposition 4 in the Appendix)

$$R(\mathrm{w}) := \mathbb{E}[\widehat{R}(\mathrm{w})] = \frac{1}{2} \sum_{i_0,i_1,i_2=1}^{d_0,d_1,d_2} \mathbf{W}_2(i_2,i_1)^2 \mathbf{W}_1(i_1,i_0)^2, \tag{3}$$

which is precisely the squared $\ell_2$ path-norm of the network (Neyshabur et al., 2015). We note that such a connection has been previously established for deep linear networks Mianjy et al. (2018); Mianjy & Arora (2019); here we have extended that result to single hidden layer ReLU networks.

Next, in order to understand the generalization properties of the dropout algorithm, we bound the Rademacher complexity of the function class with bounded expected explicit regularizer, i.e.

$$\mathcal{F}_\alpha := \{f : \mathrm{x} \mapsto f_\mathrm{w}(\mathrm{x}) : \sum_{j=1}^{d_{k-1}} \|\mathbf{W}_k(:,j)\|^2 a_j^2 \le \alpha\},$$

where $a_j^2 := \mathbb{E}_\mathrm{x}[\widehat{a}_j^2] = \mathbb{E}_\mathrm{x}[a_{j,k-1}(\mathrm{x})^2]$ is the expected squared neural activation for the $j$-th hidden node. For simplicity, we focus on networks with one output neuron; extension to multiple output neurons is rather straightforward. Recall that the Rademacher complexity is a sample dependent measure of the capacity of a hypothesis class that bounds the generalization gap, i.e., the difference between the empirical and true risks (Bartlett & Mendelson, 2002). Let $\mathcal{S} = \{(\mathrm{x}_1,y_1),\cdots,(\mathrm{x}_n,y_n)\}$ be a sample of size $n$. The empirical Rademacher complexity of $\mathcal{F}_\alpha$ with respect to $\mathcal{S}$, and the expected Rademacher complexity are defined, respectively, as:

$$\mathfrak{R}_\mathcal{S}(\mathcal{F}) = \mathbb{E}_\sigma \sup_{f\in\mathcal{F}} \frac{1}{n} \sum_{i=1}^n \sigma_i f(\mathrm{x}_i), \quad \mathfrak{R}_n(\mathcal{F}_\alpha) = \mathbb{E}_\mathrm{x}[\mathfrak{R}_\mathcal{S}(\mathcal{F}_\alpha)].$$

The following lemma gives an upper bound on the expected Rademacher complexity of $\mathcal{F}_\alpha$, which only depends on $\alpha$ and the width of the top layer $\mathrm{d}_{k-1}$.

**Lemma 1.** *For any sample size $n$ and any $\alpha > 0$ it holds that $\mathfrak{R}_n(\mathcal{F}_\alpha) \le \sqrt{\frac{d_{k-1}\alpha}{n}}$.*

We would like to remark that the expected regularizer due to dropout, i.e. $R(\mathrm{w})$, appears naturally when bounding the Rademacher complexity of deep neural networks. In particular, as we show in the proof of Lemma 1, the following upper bound holds for any class of neural networks $\mathcal{F}$:

$$\mathfrak{R}_n(\mathcal{F}) \le \frac{1}{\sqrt{n}} \sum_{j=1}^{d_{k-1}} |\mathbf{W}_k(1,j)a_j| \le \frac{1}{\sqrt{n}} \sqrt{d_{k-1} \sum_{j=1}^{d_{k-1}} \mathbf{W}_k(1,j)^2 a_j^2},$$

where the second inequality is due to Cauchy-Schwartz. In particular, dropout directly controls the summation in the right hand side above. However, note that the second inequality above can in general be very loose for large width $d_{k-1}$, specifically if a small subset of hidden nodes $\mathcal{J} \subset [d_{k-1}]$ co-adapt in a way that for all $j \in [d_{k-1}] \setminus \mathcal{J}$, the other hidden nodes are almost inactive, i.e. $\mathbf{W}_k(1,j)a_j \approx 0$. In this sense, dropout breaks "co-adaptation" between neurons by promoting solutions with (almost) equally contributing hidden neurons. For two-layer linear networks and deep linear networks with one output neuron, Mianjy et al. (2018) and Mianjy & Arora (2019), respectively, showed that the minimum of the dropout objective is only achieved by *equalized* networks, for which $|\mathbf{W}_k(1,j)a_j| = |\mathbf{W}_k(1,j')a_{j'}|, \ \forall j \in [d_{k-1}]$. For such equalized networks, the second inequality above holds with equality, where the value of the explicit regularizer gives a tight bound on the Rademacher complexity. Whether such equalization property extends beyond linear activations is an interesting open question which we leave for future work.

The main result of this section, is the following theorem that bounds the generalization gap in networks trained by dropout.

**Theorem 2.** Consider deep neural networks with ReLU activation functions. Let $\mathrm{w} = \{\mathbf{W}_i\}_{i=1}^k$ be a minimizer of the dropout objective. Assume that $\|\mathbf{W}_k\|_F \prod_{i=1}^{k-1} \|\mathbf{W}_i\| \le \sqrt{M}$, and that

$\sup_{x \in \mathcal{X}} \|x\| \leq \sqrt{B}, \sup_{y \in \mathcal{Y}} |y| \leq 1$. Let $\widehat{R}(w) \leq \alpha/2$ be an upper bound on the explicit regularizer at w. Then, with probability at least $1 - 2\delta$, over a sample of size $n$, we have that

$$L(w) \leq \widehat{L}(w) + C(1 + \sqrt{\alpha d_{k-1}})\sqrt{\frac{\alpha d_{k-1}}{n}} + C'(1 + \sqrt{d_{k-1}\alpha})^2 \sqrt{\frac{\log(2/\delta)}{2n}},$$

as long as $n = \Omega(B^2 M^2 \log(2/\delta)/\alpha^2)$, where $C$ and $C'$ are absolute constants.

In practice, modern over-parameterized deep neural networks are usually trained to zero empirical loss, i.e., if w is a minimizer of the dropout objective, then $\widehat{L}(w)$ is negligible (see, e.g. Figure 2 for the empirical error of convolutional neural networks trained with dropout on Fashion MNIST and CIFAR-10). In such settings, we expect the last two terms on the right hand side to be the dominant terms – this implies that the right hand side above is in the order of $O(\alpha d_{k-1}\sqrt{\log(2/\delta)/n})$. Interestingly, the upper bound presented in Theorem 2 has no explicit dependence on the network depth, except through the assumption that $\|W_k\|_F \prod_{i=1}^{k-1} \|W_i\| \leq \sqrt{M}$, which we argue is a very mild assumption and somewhat inevitable, as discussed in Golowich et al. (2018). In particular, if the spectral norm of the weight matrices are concentrated around 1, then M is very small. On the contrary, most generalization bounds on deep networks depend on the product of the *stable rank* of the layers, resulting in exponential dependence on the network depth (e.g., see Bartlett et al. (2017); Neyshabur et al. (2017) and the references therein). Furthermore, there is no explicit dependence on the widths of the hidden layers, except through $\sqrt{\alpha d_{k-1}}$. We note that in most state-of-the-art models, $d_{k-1}$ is not large, and as we discussed earlier, dropout implicitly equalizes the network in a way that $\sqrt{\alpha d_{k-1}} \approx \sum_{j=1}^{d_{k-1}} |W_k(1,j)a_j|$, which does not explicitly depend on the width. Finally, we remark that Theorem 2 is not limited to ReLU and can be extended to any Lipschitz activation function.

## 3.1 LINEAR REGRESSION

We recall recent results from Mianjy et al. (2018) and Mianjy & Arora (2019) that characterize the explicit regularizer $\widehat{R}(w)$ in the simpler setting of deep linear networks, where $\sigma(\cdot)$ is the identity map and the network computes a linear mapping $f_w : x \mapsto W_k \cdots W_1 x$. In particular, they showed that if $f$ is a deep linear network with one output neuron, then it holds that

$$\nu \|f\|_{\widehat{C}}^2 = \min_{f = f_w} \widehat{R}(w),$$

where $\nu$ is a regularization parameter independent of the parameters w. For linear regression (i.e., for $k = 1$ and $u = W_1^\top \in \mathbb{R}^{d_0}$), this result implies that least squares regression using dropout amounts to solving the following regularized problem:

$$\min_{u \in \mathbb{R}^{d_0}} \frac{1}{n} \sum_{i=1}^{n} (y_i - u^\top x_i)^2 + \nu \|u\|_{\widehat{C}}^2.$$

All the minimizers of the above problem are solutions to the following system of linear equations $(1 + \nu)X^\top X u = X^\top y$, where $X = [x_1, \cdots, x_n]^\top \in \mathbb{R}^{n \times d_0}, y = [y_1, \cdots, y_n]^\top \in \mathbb{R}^{n \times 1}$ are the design matrix and the response vector, respectively. Recall that if standard Tikhonov regularization with parameter $\nu$ were used instead of dropout, then the minimizer would have been the solutions to the system of linear equations $(X^\top X + \nu I)u = X^\top y$. The spectral augmentation due to Tikhonov regularization is helpful because it results in a well-posed problem, even if the the original ($\nu = 0$) system was under-determined. On the contrary, the dropout regularizer manifests itself merely as a scaling of the parameters, under which the problem remains ill-posed e.g. if $n < d_0$. More importantly, Tikhonov regularization discards the directions that account for small variance in data even when they exhibit good discriminability, which is a useful prior for learning in general. Dropout, however, does not seem to yield a useful inductive bias for linear regression. However, in the case of deep neural networks, as we show in the previous section, dropout does yield a useful inductive bias. In the rest of the paper, we confirm this theoretical result with an extensive empirical study.

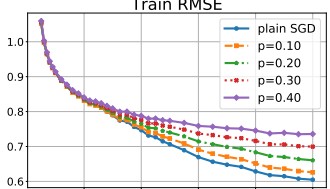 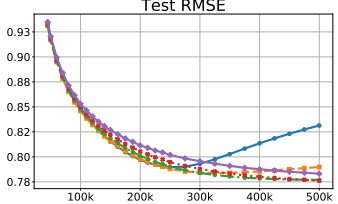 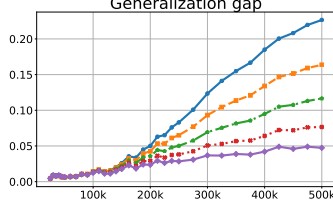

Figure 1: MovieLens dataset: the training error (**left**), the test error (**middle**), and the generalization gap (**right**) for plain SGD and dropout with $p \in \{0.1, 0.2, 0.3, 0.4\}$ as a function of the number of iterations. The factorization size is $d_1 = 70$.

| | plain SGD | | dropout | | | |
|---|---|---|---|---|---|---|
| **width** | last iterate | best iterate | $p = 0.1$ | $p = 0.2$ | $p = 0.3$ | $p = 0.4$ |
| $d_1 = 30$ | 0.8041 | 0.7938 | 0.7805 | 0.785 | 0.7991 | 0.8186 |
| $d_1 = 70$ | 0.8315 | 0.7897 | 0.7899 | 0.7771 | 0.7763 | 0.7833 |
| $d_1 = 110$ | 0.8431 | 0.7873 | 0.7988 | 0.7813 | 0.7742 | 0.7743 |
| $d_1 = 150$ | 0.8472 | 0.7858 | 0.8042 | 0.7852 | 0.7756 | 0.7722 |
| $d_1 = 190$ | 0.8473 | 0.7844 | 0.8069 | 0.7879 | 0.7772 | 0.772 |

Table 1: MovieLens dataset: Test RMSE of plain SGD as well as the dropout algorithm with various dropout rates for various factorization sizes. The grey cells shows the best performance(s) in each row.

## 4 EXPERIMENTAL RESULTS

In this section, we empirically evaluate our theoretical findings on several real world datasets[1]. All results are averaged over 50 independent runs with random initialization. We report experiments for plain SGD as well as dropout with various dropout rates. For deep neural networks, the activation function is always ReLU.

### 4.1 MATRIX COMPLETION

In this section, we explore the generalization properties of matrix completion with dropout algorithm on the MovieLens dataset Harper & Konstan (2016). MovieLens is a publicly available collaborative filtering dataset that contains 10M ratings for 11K movies by 72K users of the online movie recommender service MovieLens.

We initialize the factors using standard He initialization scheme. We train the model for 100 epochs over the training data, where we use a fixed learning rate of `lr = 1`, and a batch size of 2000. In our experiments, changing the learning rate or the batch size does not significantly improve the performance of any of these algorithms. We report the results for plain SGD ($p = 0.0$) as well as the dropout algorithm with $p \in \{0.1, 0.2, 0.3, 0.4\}$.

Figure 1 shows the progress in terms of the training and test error as well as the gap between them as a function of the number of iterations for $d_1 = 70$. It can be seen that plain SGD is the fastest in minimizing the empirical risk. The dropout rate clearly determines the trade-off between the approximation error and the estimation error: as the dropout rate $p$ increases, the algorithm favors less complex solutions that suffer larger empirical error (left figure) but enjoy smaller generalization gap (right figure). The best trade-off here seems to be achieved by a moderate dropout rate of $p = 0.3$. We observe similar behaviour for different factorization sizes; please see the Appendix for additional plots with factorization sizes $d_1 \in \{30, 110, 150, 190\}$.

It is remarkable, how even in the "simple" problem of matrix completion, plain SGD lacks a proper inductive bias. As it is clearly depicted in the middle plot, without *explicit* regularization – in particular early stopping or dropout in this figure – SGD suffers from gross overfitting. We further illustrate this fact in Table 1, where we compare the test root-mean-squared-error (RMSE) of plain SGD with the dropout algorithm, for various factorization sizes. To show the superiority of dropout over SGD with early stopping, we give SGD the advantage of having access to the *test set* (and not

---

[1]To ensure reproducibility of the results, we have uploaded the code.

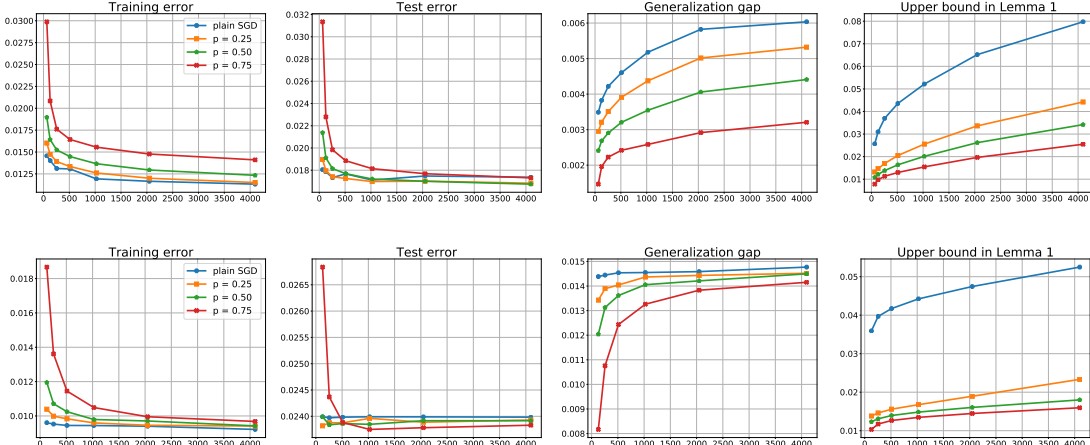

Figure 2: Fashion MNIST (**top**) and CIFAR-10 (**bottom**): the training error, the generalization gap, and the Rademacher complexity bound in Lemma 1 for plain SGD and dropout with $p \in \{0.25, 0.50, 0.75\}$ as a function of the width of the top hidden layer.

a separate validation set), and report the best iterate in the third column. Even with this impractical privilege, dropout performs significantly better ($> 0.01$ difference in test RMSE).

## 4.2 DEEP NEURAL NETWORKS

In this section, we report our experimental results for training convolutional neural networks with and without dropout, on Fashion MNIST and CIFAR-10. The Fashion MNIST dataset of Zalando's article images contains 60K training examples and 10K test examples each, where each example is a $28 \times 28$ grayscale image, associated with a label from 10 classes. The CIFAR-10 dataset consists of 60K $32 \times 32$ color images in 10 classes, with 6k images per class, divided into a training set and a test set of sizes 50K and 10K, respectively Krizhevsky et al. (2009).

For the Fashion MNIST dataset, we use a convolutional neural network with one convolutional layer and two fully connected layers. The convolutional layer has 16 filters, padding and stride of 2, and kernel size of 5. We report experiments on networks with the width of the top hidden layer chosen from `width` $\in \{2^5, 2^6, \cdots 2^{12}\}$. In all experiments, the linear layer weights are initialized using standard Xavier initialization, and a fixed learning rate `lr` $= 0.5$ and a mini-batch of size 256 is used to perform the updates. We train the models for 30 epochs over the whole training set.

For CIFAR-10, we use an AlexNet (Krizhevsky et al., 2012), where the layers are modified accordingly to match the dataset. The only difference here is that we apply dropout to the top hidden layer, whereas in Krizhevsky et al. (2012), dropout is used on the second and the third hidden layers from the top. We report experiments on networks with the width of the top hidden layer chosen from `width` $\in \{2^5, 2^6, \ldots, 2^{12}\}$. In all the experiments, an initial learning rate `lr` $= 5$ and a mini-batch of size 256 is used to perform the updates. We train the models for 100 epochs over the whole training set. We decay the learning rate by a factor of 10 every 30 epochs. We did not observe significant changes in the training/test errors by running the experiments longer.

Figure 2 shows the test error, training error, the generalization gap, and the Rademacher complexity bound in Lemma 1 as a function of the number of hidden nodes in the top hidden layer. It is not at all surprising that on all datasets, the training error of plain SGD is always the smallest, whereas dropout always enjoys a smaller generalization gap. For Fashion MNIST, which is a fairly "simple" dataset, there is no significant difference between the test performance of plain SGD and dropout; dropout performs better only when the network is highly over-parameterized. However, for CIFAR-10, dropout is always helpful in achieving better test performance. More interestingly, with dropout, the bound on the Rademacher complexity is predictive of the generalization gap, in the sense that 1) both are roughly in the same range ($\approx 0.015$), and 2) a smaller bound corresponds to a curve with smaller generalization gap.

## 5 DISCUSSION

Motivated by the success of dropout in deep learning, we propose a dropout algorithm for matrix sensing and show that it enjoys strong generalization guarantees as well as competitive test performance on the MovieLens dataset. We then focus on deep regression under the squared loss and show that the regularizer due to dropout serves as a strong complexity measure for the underlying class of deep neural networks, using which we give a generalization error bound in terms of the value of the regularizer. We evaluate our theoretical findings for training convolutional neural networks on real world datasets including CIFAR-10.

We leave several important questions for future work. It would be interesting to study the optimization theoretic aspects of training with dropout. In particular, the results presented in this paper is based on the assumption that dropout finds an (approximate) empirical minimizer; although this is widely supported by practice, giving precise convergence rates remains an open question. Another direction for future work is to understand the function class approximated by neural networks with dropout layer. For example, it is known that deep ReLU networks represent the class of piecewise linear functions (Arora et al., 2018). What can we say about the function class approximated by neural networks with norm-bounded weight matrices, due to dropout.

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

# Supplementary Materials for
# "Dropout: Explicit Forms and Capacity Control"

## A    AUXILIARY RESULTS

**Theorem 3** (Hoeffding's inequality: Theorem 2.6.2 Vershynin (2018)). *Let $X_1, \ldots, X_N$ be independent, mean zero, sub-Gaussian random variables. Then, for every $t \geq 0$, we have*

$$\mathbb{P}\left(\left|\frac{1}{N}\sum_{i=1}^{N}X_i\right| \geq t\right) \leq 2e^{-\frac{ct^2N^2}{\sum_{i=1}^{N}\|X_i\|_{\psi_2}^2}}$$

**Theorem 4** (Theorem 10.3 of Mohri et al. (2018)). *Assume that $\|h - f\|_\infty \leq M$ for all $h \in \mathcal{H}$. Then, for any $\delta > 0$, with probability at least $1 - \delta$ over a sample $\{(x_i, y_i), \ i \in [n]\}$ of size $n$, the following inequalities holds uniformly for all $h \in \mathcal{H}$.*

$$\mathbb{E}[|h(x) - f(x)|^2] \leq \frac{1}{n}\sum_{i=1}^{n}|h(x_i) - f(x_i)|^2 + 4M\mathfrak{R}_n(\mathcal{H}) + M^2\sqrt{\frac{\log(2/\delta)}{2n}}$$

**Theorem 5** (Theorem 3.3 in Mianjy et al. (2018)). *For any pair of matrices $U \in \mathbb{R}^{d_2 \times d_1}, V \in \mathbb{R}^{d_0 \times d_1}$, there exist a rotation matrix $Q \in SO(d_1)$ such that rotated matrices $\tilde{U} := UQ, \tilde{V} := VQ$ satisfy $\|\tilde{u}_i\|\|\tilde{v}_i\| = \frac{1}{d_1}\|UV^\top\|_*$, for all $i \in [d_1]$.*

**Theorem 6** (Theorem 1 in Foygel et al. (2011)). *Assume that $p(i)q(j) \geq \frac{\log(d_2)}{n\sqrt{d_2 d_0}}$ for all $i \in [d_2], j \in [d_0]$. For any $\alpha > 0$, let $\mathcal{M}_\alpha := \{M \in \mathbb{R}^{d_2 \times d_1} : \|\operatorname{diag}(\sqrt{p})M\operatorname{diag}(\sqrt{q})\|_*^2 \leq \alpha\}$ be the class of linear transformations with weighted trace-norm bounded with $\sqrt{\alpha}$. Then the expected Rademacher complexity of $\mathcal{M}_\alpha$ is bounded as follows:*

$$\mathfrak{R}_n(\mathcal{M}_\alpha) \leq O\left(\sqrt{\frac{\alpha d_2 \log(d_2)}{n}}\right)$$

## B    MATRIX SENSING

**Proposition 1** (Dropout regularizer in matrix sensing). *The following holds for any $p \in [0, 1)$:*

$$\widehat{L}_{drop}(U, V) = \widehat{L}(U, V) + \lambda\widehat{R}(U, V), \quad where \ \widehat{R}(U, V) = \sum_{i=1}^{d_1}\frac{1}{n}\sum_{j=1}^{n}(u_i^\top A^{(j)}v_i)^2. \tag{4}$$

*where $\lambda = \frac{p}{1-p}$ is the regularization parameter.*

*Proof of Proposition 1.* The following equality follows from the definition of variance:

$$\mathbb{E}_b[(y_i - \langle UBV^\top, A^{(i)}\rangle)^2] = \left(\mathbb{E}_b[y_i - \langle UBV^\top, A^{(i)}\rangle]\right)^2 + \operatorname{Var}(y_i - \langle UBV^\top, A^{(i)}\rangle)$$

Recall that for a Bernoulli random variable $B_{ii}$, we have $\mathbb{E}[B_{ii}] = 1$ and $\operatorname{Var}(B_{ii}) = \frac{p}{1-p}$. Thus, the first term on right hand side is equal to $(y_i - \langle UV^\top, A^{(i)}\rangle)^2$. For the second term we have

$$\operatorname{Var}(y_i - \langle UBV^\top, A^{(i)}\rangle) = \operatorname{Var}(\sum_{j=1}^{d_1}B_{jj}u_j^\top A^{(i)}v_j) = \sum_{j=1}^{d_1}(u_j^\top A^{(i)}v_j)^2 \operatorname{Var}(B_{jj})$$

$$= \frac{p}{1-p}\sum_{j=1}^{d_1}(u_j^\top A^{(i)}v_j)^2$$

Plugging the above into Equation (5) and averaging over samples we get

$$
\begin{aligned}
\widehat{L}_{\mathrm{drop}}(\mathrm{U}, \mathrm{V}) &= \frac{1}{n} \sum_{i=1}^{n} \mathbb{E}_{\mathbf{b}}[(y_i - \langle \mathrm{UBV}^\top, A^{(i)} \rangle)^2] \\
&= \frac{1}{n} \sum_{i=1}^{n} (y_i - \langle \mathrm{UV}^\top, A^{(i)} \rangle)^2 + \frac{1}{n} \sum_{i=1}^{n} \frac{p}{1-p} \sum_{j=1}^{d_1} (\mathbf{u}_j^\top A^{(i)} \mathbf{v}_j)^2 \\
&= \widehat{L}(\mathrm{U}, \mathrm{V}) + \frac{p}{1-p} \widehat{R}(\mathrm{U}, \mathrm{V}).
\end{aligned}
$$

which completes the proof. $\qquad\square$

**Proposition 2.** *[Induced regularizer] The followings hold true.*

1. **Matrix completion.** *For $j \in [n]$, let $A^{(j)}$ be an indicator matrix whose $(i, k)$-th element is selected randomly with probability $p(i)q(k)$, where $p(i)$ and $q(k)$ denote the probability of choosing the $i$-th row and the $k$-th column. Then $\Theta(M) = \frac{1}{d_1} \| \operatorname{diag}(\sqrt{p}) U V^\top \operatorname{diag}(\sqrt{q}) \|_*^2$.*

2. **i.i.d. measurements.** *For all $j \in [n]$, let the elements of $A^{(j)}$ be distributed i.i.d. with zero mean and unit variance. Then $\Theta(M) = \frac{1}{d_1} \|M\|_*^2$.*

*Proof of Proposition 2.* For any pair of factors $(\mathrm{U}, \mathrm{V})$ it holds that

$$
\begin{aligned}
R(\mathrm{U}, \mathrm{V}) &= \sum_{i=1}^{d_1} \mathbb{E}(\mathbf{u}_i^\top A \mathbf{v}_i)^2 \\
&= \sum_{i=1}^{d_1} \sum_{j=1}^{d_2} \sum_{k=1}^{d_0} p(j)q(k)(\mathbf{u}_i^\top \mathbf{e}_j \mathbf{e}_k^\top \mathbf{v}_i)^2 \\
&= \sum_{i=1}^{d_1} \sum_{j=1}^{d_2} \sum_{k=1}^{d_0} p(j)q(k) \mathrm{U}(j,i)^2 \mathrm{V}(k,i)^2 \\
&= \sum_{i=1}^{d_1} \| \operatorname{diag}(\sqrt{p}) \mathbf{u}_i \|^2 \| \operatorname{diag}(\sqrt{q}) \mathbf{v}_i \|^2 \\
&\geq \frac{1}{d_1} \left( \sum_{i=1}^{d_1} \| \operatorname{diag}(\sqrt{p}) \mathbf{u}_i \| \| \operatorname{diag}(\sqrt{q}) \mathbf{v}_i \| \right)^2 \\
&= \frac{1}{d_1} \left( \sum_{i=1}^{d_1} \| \operatorname{diag}(\sqrt{p}) \mathbf{u}_i \mathbf{v}_i^\top \operatorname{diag}(\sqrt{q}) \|_* \right)^2 \\
&\geq \frac{1}{d_1} \left( \| \operatorname{diag}(\sqrt{p}) \sum_{i=1}^{d_1} \mathbf{u}_i \mathbf{v}_i^\top \operatorname{diag}(\sqrt{q}) \|_* \right)^2 \\
&= \frac{1}{d_1} \| \operatorname{diag}(\sqrt{p}) \mathrm{U} \mathrm{V}^\top \operatorname{diag}(\sqrt{q}) \|_*^2
\end{aligned}
$$

where the first inequality is due to Cauchy-Schwartz and the second inequality follows from the triangle inequality. The equality right after the first inequality follows from the fact that for any two vectors $\mathbf{a}, \mathbf{b}$, $\| \mathbf{a}\mathbf{b}^\top \|_* = \| \mathbf{a}\mathbf{b}^\top \| = \|\mathbf{a}\|\|\mathbf{b}\|$. Since the inequalities hold for any $\mathrm{U}, \mathrm{V}$, it implies that

$$
\Theta(\mathrm{U}\mathrm{V}^\top) \geq \frac{1}{d_1} \| \operatorname{diag}(\sqrt{p}) \mathrm{U} \mathrm{V}^\top \operatorname{diag}(\sqrt{q}) \|_*^2.
$$

Applying Theorem 5 on $(\operatorname{diag}(\sqrt{p})\mathrm{U}, \operatorname{diag}(\sqrt{p})\mathrm{V})$, there exist a rotation matrix $\mathrm{Q}$ such that

$$
\| \operatorname{diag}(\sqrt{p}) \mathrm{U} \mathbf{q}_i \| \| \operatorname{diag}(\sqrt{q}) \mathrm{V} \mathbf{q}_i \| = \frac{1}{d_1} \| \operatorname{diag}(\sqrt{p}) \mathrm{U} \mathrm{V}^\top \operatorname{diag}(\sqrt{q}) \|_*
$$

We evaluate the expected dropout regularizer at UQ, VQ:

$$R(\mathbf{UQ}, \mathbf{VQ}) = \sum_{i=1}^{d_1} \| \operatorname{diag}(\sqrt{p})\mathbf{Uq}_i \|^2 \| \operatorname{diag}(\sqrt{q})\mathbf{Vq}_i \|^2$$

$$= \sum_{i=1}^{d_1} \frac{1}{d_1^2} \| \operatorname{diag}(\sqrt{p})\mathbf{UV}^\top \operatorname{diag}(\sqrt{q}) \|_*^2$$

$$= \frac{1}{d_1} \| \operatorname{diag}(\sqrt{p})\mathbf{UV}^\top \operatorname{diag}(\sqrt{q}) \|_*^2$$

$$\leq \Theta(\mathbf{UV}^\top)$$

which completes the proof of the first part.

Similarly for the second part, we first show that any pair of factors $(\mathbf{U}, \mathbf{V})$, $R(\mathbf{U}, \mathbf{V}) geq \frac{1}{d_1} \|\mathbf{UV}^\top\|_*^2$:

$$R(\mathbf{U}, \mathbf{V}) = \sum_{i=1}^{d_1} \mathbb{E}(\mathbf{u}_i^\top \mathbf{A} \mathbf{v}_i)^2$$

$$= \sum_{i=1}^{d_1} \mathbb{E}(\sum_{j=1}^{d_2} \sum_{j=1}^{d_2} \mathbf{U}_{ji} \mathbf{A}_{jk} \mathbf{V}_{ki})^2$$

$$= \sum_{i=1}^{d_1} \sum_{j,j'=1}^{d_2} \sum_{k,k'=1}^{d_0} \mathbf{U}_{ji} \mathbf{U}_{j'i} \mathbf{V}_{ki} \mathbf{V}_{k'i} \mathbb{E}[\mathbf{A}_{jk} \mathbf{A}_{j'k'}]$$

$$= \sum_{i=1}^{d_1} \sum_{j=1}^{d_2} \sum_{k=1}^{d_0} \mathbf{U}_{ji}^2 \mathbf{V}_{ki}^2 \mathbb{E}[\mathbf{A}_{jk}^2]$$

$$= \sum_{i=1}^{d_1} \sum_{j=1}^{d_2} \sum_{k=1}^{d_0} \mathbf{U}_{ji}^2 \mathbf{V}_{ki}^2$$

$$= \sum_{i=1}^{d_1} \|\mathbf{u}_i\|^2 \|\mathbf{v}_i\|^2$$

$$\geq \frac{1}{d_1} \left( \sum_{i=1}^{d_1} \|\mathbf{u}_i\| \|\mathbf{v}_i\| \right)^2$$

$$= \frac{1}{d_1} \left( \sum_{i=1}^{d_1} \|\mathbf{u}_i \mathbf{v}_i^\top\|_* \right)^2$$

$$\geq \frac{1}{d_1} \left( \| \sum_{i=1}^{d_1} \mathbf{u}_i \mathbf{v}_i^\top \|_* \right)^2 = \frac{1}{d_1} \|\mathbf{UV}^\top\|_*^2$$

where the first and the second inequaliteis are due to Cauchy-Schwartz and the triangle inequality, respectively. The equality right after the first inequality follows because for any pair of vectors $\mathbf{a}, \mathbf{b}$, it holds that $\|\mathbf{ab}^\top\|_* = \|\mathbf{ab}^\top\| = \|\mathbf{a}\|\|\mathbf{b}\|$. Now again using Theorem 5 on $(\mathbf{U}, \mathbf{V})$, there exist a rotation matrix Q such that $\|\mathbf{Uq}_i\|\|\mathbf{Vq}_i\| = \frac{1}{d_1}\|\mathbf{UV}^\top\|_*$. We evaluate the expected dropout regularizer at UQ, VQ:

$$R(\mathbf{UQ}, \mathbf{VQ}) = \sum_{i=1}^{d_1} \|\mathbf{Uq}_i\|^2 \|\mathbf{Vq}_i\|^2 = \sum_{i=1}^{d_1} \frac{1}{d_1^2} \|\mathbf{UV}^\top\|_*^2 = \frac{1}{d_1} \|\mathbf{UV}^\top\|_*^2 \leq \Theta(\mathbf{UV}^\top)$$

which completes the proof of the the second part. $\qquad \square$

**Lemma 2.** *Assume $U, V$ is such that $\max_i \|U(i,:)\|^2 \leq \gamma$, $\max_i \|V(i,:)\|^2 \leq \gamma$. Then, with probability at least $1 - \delta$ over a sample of size $n$, we have that*

$$|R(U, V) - \widehat{R}(U, V)| \leq \frac{C\gamma^2 \sqrt{\log(2/\delta)}}{\sqrt{n}}.$$

*Proof of Lemma 2.* Define $X_\ell := \sum_{w=1}^{d_1} (u_w^\top A^{(\ell)} v_w)^2$ and observe that

$$
\begin{aligned}
X_\ell &= \sum_{w=1}^{d_1} \left( \sum_{i,j} U_{iw} V_{jw} A_{ij}^{(\ell)} \right)^2 \\
&= \sum_{w=1}^{d_1} \sum_{i,i',j,j'} U_{iw} U_{i'w} V_{jw} V_{j'w} A_{ij}^{(\ell)} A_{i'j'}^{(\ell)} \\
&= \sum_{w=1}^{d_1} \sum_{i,j} U_{iw}^2 V_{jw}^2 A_{ij}^{(\ell)} \\
&\leq \max_{i,j} \sum_{w=1}^{d_1} U_{iw}^2 V_{jw}^2 \\
&\leq \max_{i,j} \|U(i,:)\|^2 \|V(j,:)\|^2 \leq \gamma^2
\end{aligned}
$$

where the third equality follows because for an indicator matrix $A^{(\ell)}$, it holds that $A_{ij}^{(\ell)} A_{i'j'}^{(\ell)} = 0$ if $(i,j) \neq (i',j')$. Thus, $X_{w,\ell}$ is a sub-Gaussian (more strongly, bounded) random variable with mean $\mathbb{E}[X_\ell] = R(U,V)$ and sub-Gaussian norm $\|X_\ell\|_{\psi_2} \leq \gamma^2/\ln(2)$. Furthermore, $\|X_\ell - R(U,V)\|_{\psi_2} \leq C'\|X_\ell\|_{\psi_2} \leq C\gamma^2$, for some absolute constants $C', C$ (Lemma 2.6.8 of Vershynin (2018)). Using Theorem 3, for $t = Cd_1\sqrt{\frac{\log 2/\delta}{n}}$ we get that:

$$
\mathbb{P}\left( \left| \widehat{R}(U,V) - R(U,V) \right| \geq t \right) = \mathbb{P}\left( \left| \frac{1}{n} \sum_{\ell=1}^n X_\ell - R(U,V) \right| \geq C\gamma^2 \sqrt{\frac{\log 2/\delta}{n}} \right) \leq \delta
$$

which completes the proof. $\qquad\square$

*Proof of Theorem 1.* We use Theorem 4 to give a bound on the generalization gap. From Lemma 2, we have with probability at least $1 - \delta$ that

$$
\begin{aligned}
\frac{1}{d_1} \| \operatorname{diag}(\sqrt{p}) U V^\top \operatorname{diag}(\sqrt{q}) \|_*^2 = \Theta(UV^\top) &\leq R(U,V) && \text{(definition of } \Theta(\cdot)) \\
&\leq \widehat{R}(U,V) + \frac{C\gamma^2 \sqrt{\log(2/\delta)}}{\sqrt{n}} && \text{(Lemma 2)} \\
&\leq \frac{\alpha}{2d_1} + \frac{C\gamma^2 \sqrt{\log(2/\delta)}}{\sqrt{n}} && \text{(assumption of the Theorem)} \\
&\leq \frac{\alpha}{d_1} && \text{(whenever } n \geq 4d_1^2 \gamma^4 C^2 \log(2/\delta)/\alpha^2)
\end{aligned}
$$

Define the class of predictors with weighted trace-norm bounded by $\sqrt{\alpha}$, i.e. $\mathcal{M}_\alpha = \{M : \|\operatorname{diag}(\sqrt{p}) M \operatorname{diag}(\sqrt{q})\|_*^2 \leq \alpha\}$. By Theorem 6, we have that $\mathfrak{R}_n(\mathcal{F}_\alpha) \leq \sqrt{\frac{\alpha d_2 \log(d_2)}{n}}$. It remains to bound the supremum deviation between elements of $M_*$ and $UV^\top \in \mathcal{M}_\alpha$:

$$
\begin{aligned}
\max_A |\langle M_* - UV^\top, A \rangle| &= \max_{i,j} |\langle M_* - UV^\top, e_i e_j^\top \rangle| \\
&\leq \max_{i,j} |M_*(i,j)| + \max_{i,j} |\langle UV^\top, e_i e_j^\top \rangle| \\
&\leq \|M_*\| + \max_{i,j} |\langle U(i,:), V(j,:) \rangle| \\
&\leq 1 + \max_{i,j} \|U(i,:)\| \|V(j,:)\| \leq 1 + \gamma
\end{aligned}
$$

Plugging the above results in Theorem 4, we get

$$
L(U,V) \leq \widehat{L}(U,V) + C(1+\gamma)\sqrt{\frac{\alpha d_2 \log(d_2)}{n}} + C'(1+\gamma^2)\sqrt{\frac{\log(2/\delta)}{2n}}
$$

which completes the proof. $\qquad\square$

## C  DEEP NEURAL NETWORKS

**Proposition 3** (Dropout regularizer in deep regression)**.**

$$\widehat{L}_{drop}(w) = \widehat{L}(w) + \widehat{R}(w), \;\; where \;\; \widehat{R}(w) = \lambda \sum_{j=1}^{d_{k-1}} \|W_k(:,j)\|^2 \widehat{a}_j^2.$$

*where $\widehat{a}_j = \sqrt{\frac{1}{n} \sum_{i=1}^{n} a_{j,k-1}(x_i)^2}$ and $\lambda = \frac{p}{1-p}$ is the regularization parameter.*

*Proof of Proposition 3.* Recall that $\mathbb{E}[\mathbf{B}_{ii}] = 1$ and $\mathrm{Var}(\mathbf{B}_{ii}) = \frac{p}{1-p}$. Conditioned on x, y in the current mini-batch, we have that

$$\mathbb{E}_{\mathbf{B}}[\|y - W_k \mathbf{B} a_{k-1}(x)\|^2] = \sum_{i=1}^{d_k} \mathbb{E}_{\mathbf{B}}(y_i - W_k(i,:)^\top \mathbf{B} a_{k-1}(x))^2. \tag{5}$$

where $a_j(x) \in \mathbb{R}^{d_j}$ is the activation vector of the $j$-th hidden layer for input x, i.e. $a_j(x)[i] = a_{i,j}(x)$. The following holds by the definition of variance for each of the summands above:

$$\mathbb{E}_{\mathbf{B}}(y_i - W_k(i,:)^\top \mathbf{B} a_{k-1}(x))^2 = \left(\mathbb{E}_{\mathbf{B}}[y_i - W_k(i,:)^\top \mathbf{B} a_{k-1}(x)]\right)^2 + \mathrm{Var}(y_i - W_k(i,:)^\top \mathbf{B} a_{k-1}(x))$$

Since $\mathbb{E}[\mathbf{B}] = \mathbf{I}$, the first term on right hand side is equal to $(y_i - W_k(:,i)^\top a_{k-1}(x))^2$. For the second term we have

$$\mathrm{Var}(y_i - W_k(i,:)^\top \mathbf{B} a_{k-1}(x)) = \mathrm{Var}(W_k(i,:)^\top \mathbf{B} a_{k-1}(x))$$
$$= \mathrm{Var}\left(\sum_{j=1}^{d_{k-1}} W_k(i,j) \mathbf{B}_{jj} a_{j,k-1}(x)\right)$$
$$= \sum_{j=1}^{d_{k-1}} (W_k(i,j) a_{j,k-1}(x))^2 \, \mathrm{Var}(\mathbf{B}_{jj})$$
$$= \frac{p}{1-p} \sum_{j=1}^{d_{k-1}} W_k(i,j)^2 a_{j,k-1}(x)^2$$

Plugging the above into Equation (5)

$$\mathbb{E}_{\mathbf{B}}[\|y - W_k \mathbf{B} a_{k-1}(x)\|^2] = \|y - W_k a_{k-1}(x)\|^2 + \frac{p}{1-p} \sum_{j=1}^{d_{k-1}} \|W_k(:,j)\|^2 a_{j,k-1}(x)^2$$

Now taking the empirical average with respect to x, y, we get

$$\widehat{L}_{\mathrm{drop}}(\mathbf{w}) = \widehat{L}(\mathbf{w}) + \frac{p}{1-p} \sum_{j=1}^{d_{k-1}} \|W_k(:,j)\|^2 \widehat{a}_j^2 = \widehat{L}(\mathbf{w}) + \widehat{R}(\mathbf{w})$$

which completes the proof. $\qquad\square$

**Proposition 4.** *Consider a two layer neural network $f_w(\cdot)$ with ReLU activation functions in the hidden layer. Furthermore, assume that the marginal input distribution $\mathbb{P}_{\mathcal{X}}(x)$ is symmetric and isotropic, i.e., $\mathbb{P}_{\mathcal{X}}(x) = \mathbb{P}_{\mathcal{X}}(-x)$ and $\mathbb{E}[xx^\top] = I$. Then the following holds for the expected explicit regularizer due to dropout:*

$$R(w) := \mathbb{E}[\widehat{R}(w)] = \frac{\lambda}{2} \sum_{i_0,i_1,i_2=1}^{d_0,d_1,d_2} W_2(i_2,i_1)^2 W_1(i_1,i_0)^2, \tag{6}$$

*Proof of Proposition 4.* Using Proposition 3, we have that:

$$R(\mathbf{w}) = \mathbb{E}[\widehat{R}(\mathbf{w})] = \lambda \sum_{j=1}^{d_1} \|W_2(:,j)\|^2 \mathbb{E}[\sigma(W_1(j,:)^\top x)^2]$$

It remains to calculate the quantity $\mathbb{E}_{\mathrm{x}}[\sigma(\mathbf{W}_1(j,:)^\top \mathbf{x})^2]$. By symmetry assumption, we have that $\mathbb{P}_{\mathcal{X}}(\mathbf{x}) = \mathbb{P}_{\mathcal{X}}(-\mathbf{x})$. As a result, for any $\mathbf{v} \in \mathbb{R}^{d_0}$, we have that $\mathbb{P}(\mathbf{v}^\top \mathbf{x}) = \mathbb{P}(-\mathbf{v}^\top \mathbf{x})$ as well. That is, the random variable $z_j := \mathbf{W}_1(j,:)^\top \mathbf{x}$ is also symmetric about the origin. It is easy to see that $\mathbb{E}_z[\sigma(z)^2] = \frac{1}{2}\mathbb{E}_z[z^2]$.

$$\mathbb{E}_z[\sigma(z)^2] = \int_{-\infty}^{\infty} \sigma(z)^2 d\mu(z)$$
$$= \int_0^{\infty} \sigma(z)^2 d\mu(z) = \int_0^{\infty} z^2 d\mu(z)$$
$$= \frac{1}{2} \int_{\infty}^{\infty} z^2 d\mu(z) = \frac{1}{2}\mathbb{E}_z[z^2].$$

Plugging back the above identity in the expression of $R(\mathbf{w})$, we get that

$$R(\mathbf{w}) = \lambda \sum_{j=1}^{d_1} \|\mathbf{W}_2(:,j)\|^2 \mathbb{E}[(\mathbf{W}_1(j,:)^\top \mathbf{x})^2] = \lambda \sum_{j=1}^{d_1} \|\mathbf{W}_2(:,j)\|^2 \|\mathbf{W}_1(j,:)\|^2$$

where the second equality follows from the assumption that the distribution is isotropic. $\qquad \square$

*Proof of Lemma 1.* Given a dataset $\mathcal{S}$, we start by bounding the empirical Rademahcer complexity:

$$\mathfrak{R}_{\mathcal{S}}(\mathcal{F}_\alpha) = \mathbb{E}_\sigma \sup_{f_{\mathbf{w}} \in \mathcal{F}_\alpha} \frac{1}{n} \sum_{i=1}^n \sigma_i f_{\mathbf{w}}(\mathbf{x}_i)$$
$$= \mathbb{E}_\sigma \sup_{f \in \mathcal{F}_\alpha} \frac{1}{n} \sum_{i=1}^n \sigma_i \sum_{j=1}^{d_{k-1}} \mathbf{W}_k(1,j)\mathbf{a}_{j,k-1}(\mathbf{x}_i)$$
$$= \mathbb{E}_\sigma \sup_{f_{\mathbf{w}} \in \mathcal{F}_\alpha} \frac{1}{n} \sum_{j=1}^{d_{k-1}} \mathbf{W}_k(1,j)a_j \sum_{i=1}^n \sigma_i \frac{a_{j,k-1}(\mathbf{x}_i)}{a_j}$$
$$\leq \sup_{f_{\mathbf{w}} \in \mathcal{F}_\alpha} \sum_{j=1}^{d_{k-1}} |\mathbf{W}_k(1,j)a_j| \cdot \frac{1}{n} \mathbb{E}_\sigma \sup_{f_{\mathbf{w}} \in \mathcal{F}_\alpha} \max_{j \in [d_{k-1}]} |\sum_{i=1}^n \sigma_i \frac{a_{j,k-1}(\mathbf{x}_i)}{a_j}|$$

The first term on the right hand side is bounded as:

$$\sum_{j=1}^{d_{k-1}} |\mathbf{W}_k(1,j)a_j| \leq \sqrt{d_{k-1}} \sqrt{\sum_{j=1}^{d_{k-1}} \mathbf{W}_k(1,j)^2 a_j^2} \qquad \text{(Cauchy-Schwartz)}$$
$$\leq \sqrt{d_{k-1}}\alpha$$

We now consider the function class

$$\tilde{\mathcal{F}} := \{h_{\tilde{\mathbf{w}}} : \mathbf{x} \mapsto \sigma(\mathbf{v}\sigma(\mathbf{W}_{k-2}\sigma(\cdots \mathbf{W}_2\sigma(\mathbf{W}_1\mathbf{x})\cdots))), \ \mathbb{E}_{\mathbf{x}}[h_{\tilde{\mathbf{w}}}(\mathbf{x})^2] = 1\}$$

Plugging back the above and noting that $a_j \geq 0$, we have

$$\mathfrak{R}_{\mathcal{S}}(\mathcal{F}_\alpha) \leq \frac{\sqrt{d_{k-1}}\alpha}{n} \mathbb{E}_\sigma \sup_{a_j^2 = 1} \max_{j \in [d_{k-1}]} |\sum_{i=1}^n \sigma_i a_{j,k-1}(\mathbf{x}_i)|$$
$$\leq \frac{\sqrt{d_{k-1}}\alpha}{n} \mathbb{E}_\sigma \sup_{h_{\tilde{\mathbf{w}}} \in \tilde{\mathcal{F}}} |\sum_{i=1}^n \sigma_i h_{\tilde{\mathbf{w}}}(\mathbf{x}_i)|$$

Let $\tilde{\mathbf{w}}_*$ be the maximizer of the right hand side, i.e.

$$\mathbb{E}_\sigma |\sum_{i=1}^n \sigma_i h_{\tilde{\mathbf{w}}_*}(\mathbf{x}_i)| = \sup_{h_{\tilde{\mathbf{w}}} \in \tilde{\mathcal{F}}} |\sum_{i=1}^n \sigma_i h_{\tilde{\mathbf{w}}_*}(\mathbf{x}_i)|$$

Then, it holds that

$$
\begin{aligned}
\Re_n(\mathcal{F}_\alpha) = \mathbb{E}_{\mathrm{x}}[\Re_{\mathcal{S}}(\mathcal{F}_\alpha)] \\
\leq \frac{\sqrt{d_{k-1}\alpha}}{n} \mathbb{E}_{\mathrm{x},\sigma} | \sum_{i=1}^{n} \sigma_i h_{\tilde{\mathrm{w}}_*}(\mathrm{x}_i) | \\
\leq \frac{\sqrt{d_{k-1}\alpha}}{n} \sqrt{\mathbb{E}_{\mathrm{x}}\mathbb{E}_\sigma \left( \sum_{i=1}^{n} \sigma_i h_{\tilde{\mathrm{w}}_*}(\mathrm{x}_i) \right)^2} \\
= \frac{\sqrt{d_{k-1}\alpha}}{n} \sqrt{\sum_{i=1}^{n} \mathbb{E}_{\mathrm{x}} h_{\tilde{\mathrm{w}}_*}(\mathrm{x}_i)^2} = \sqrt{\frac{d_{k-1}\alpha}{n}}
\end{aligned}
$$

which completes the proof of the Lemma. $\qquad\square$

**Lemma 3.** *Assume that $\|W_k\|_F \prod_{i=1}^{k-1} \|W_i\| \leq \sqrt{M}$, and that $\sup_{x \in \mathcal{X}} \|x\| \leq \sqrt{B}, \sup_{y \in \mathcal{Y}} |y| \leq 1$. Then with probability at least $1 - \delta$ over a sample of size $n$, we have that*

$$
|R(w) - \widehat{R}(w)| \leq \frac{CBM\sqrt{\log(2/\delta)}}{\sqrt{n}},
$$

*where $C$ is an absolute constant.*

*Proof of Lemma 3.* Define $X_\ell := \sum_{j=1}^{d_{k-1}} \|\mathrm{W}_k(:,j)\|^2 a_{j,k-1}(\mathrm{x}_\ell)^2$ and observe that

$$
\begin{aligned}
X_\ell &= \sum_{j=1}^{d_{k-1}} \|\mathrm{W}_k(:,j)\|^2 a_{j,k-1}(\mathrm{x}_\ell)^2 \\
&= \sum_{j=1}^{d_{k-1}} \|\mathrm{W}_k(:,j)\|^2 \sigma \left( \mathrm{W}_{k-1}(j,:)a_{k-1}(\mathrm{x}_\ell) \right)^2 \\
&\leq \sum_{j=1}^{d_{k-1}} \|\mathrm{W}_k(:,j)\|^2 \left( \mathrm{W}_{k-1}(j,:)a_{k-1}(\mathrm{x}_\ell) \right)^2 \\
&\leq \sum_{j=1}^{d_{k-1}} \|\mathrm{W}_k(:,j)\|^2 \|\mathrm{W}_{k-1}(j,:)\|^2 \|a_{k-1}(\mathrm{x}_\ell)\|^2 \\
&\leq \|a_{k-1}(\mathrm{x}_\ell)\|^2 \max_i \|\mathrm{W}_{k-1}(i,:)\|^2 \sum_{j=1}^{d_{k-1}} \|\mathrm{W}_k(:,j)\|^2 \\
&\leq \|\mathrm{x}_\ell\|^2 \|\mathrm{W}_1\|^2 \cdots \|\mathrm{W}_{k-1}\|^2 \|\mathrm{W}_k\|_F^2 \\
&\leq BM
\end{aligned}
$$

where the first inequality follows from the definition of ReLU, and the second inequality is due to Cauchy-Schwartz. Thus, $X_{w,\ell}$ is a sub-Gaussian (more strongly, bounded) random variable with mean $\mathbb{E}[X_\ell] = R(w)$ and sub-Gaussian norm $\|X_\ell\|_{\psi_2} \leq BM/\ln(2)$. Furthermore, $\|X_\ell - R(w)\|_{\psi_2} \leq C'\|X_\ell\|_{\psi_2} \leq CBM$, for some absolute constants $C', C$ (Lemma 2.6.8 of Vershynin (2018)). Using Theorem 3, for $t = CBM\sqrt{\frac{\log 2/\delta}{n}}$ we get that:

$$
\mathbb{P}\left( \left| \widehat{R}(w) - R(w) \right| \geq t \right) = \mathbb{P}\left( \left| \frac{1}{n}\sum_{\ell=1}^{n} X_\ell - R(w) \right| \geq CBM\sqrt{\frac{\log n}{n}} \right) \leq \delta,
$$

which completes the proof. $\qquad\square$

*Proof of Theorem 2.* We use Theorem 4 to give a bound on the generalization gap. From Lemma 3, we have with probability at least $1 - \delta$ that

$$
R(\mathrm{w}) \leq \widehat{R}(\mathrm{w}) + \frac{CBM\sqrt{\log(2/\delta)}}{\sqrt{n}} \leq \frac{\alpha}{2} + \frac{\alpha}{2} = \alpha
$$

where the second inequality holds whenever $n \geq 4C^2 B^2 M^2 \log(2/\delta)/\alpha^2$. Define the class of deep neural networks with dropout regularizer bounded by $\sqrt{\alpha}$, i.e. $\mathcal{F}_\alpha = \{f_{\mathrm{w}} : R(\mathrm{w}) \leq \alpha\}$. By Lemma 1, we have that $\mathfrak{R}_n(\mathcal{F}_\alpha) \leq \sqrt{\frac{\alpha d_{k-1}}{n}}$. It remains to bound the supremum deviation between true labels and those predicted by networks that belong to $\mathcal{F}_\alpha$:

$$
\sup_{\mathrm{x,y}\sim\mathcal{D},\ f_{\mathrm{w}}\in\mathcal{F}_\alpha} \|\mathrm{y} - f_{\mathrm{w}}(\mathrm{x})\| \leq \sup_{\mathrm{x,y}\sim\mathcal{D}} \|\mathrm{y}\| + \sup_{\mathrm{x,y}\sim\mathcal{D},\ f_{\mathrm{w}}\in\mathcal{F}_\alpha} \|f_{\mathrm{w}}(\mathrm{x})\|
$$

$$
\leq 1 + \sup_{\mathrm{x,y}\sim\mathcal{D},\ f_{\mathrm{w}}\in\mathcal{F}_\alpha} \| \sum_{j=1}^{d_{k-1}} \mathrm{W}_k(:,j)\mathrm{a}_{j,k-1}(\mathrm{x})\|
$$

$$
\leq 1 + \sup_{\mathrm{x,y}\sim\mathcal{D},\ f_{\mathrm{w}}\in\mathcal{F}_\alpha} \sum_{j=1}^{d_{k-1}} \|\mathrm{W}_k(:,j)\mathrm{a}_{j,k-1}(\mathrm{x})\|
$$

$$
\leq 1 + \sup_{\mathrm{x,y}\sim\mathcal{D},\ f_{\mathrm{w}}\in\mathcal{F}_\alpha} \sqrt{d_{k-1}\sum_{j=1}^{d_{k-1}} \|\mathrm{W}_k(:,j)\|^2 \mathrm{a}_{j,k-1}(\mathrm{x})^2}
$$

$$
\leq 1 + \sqrt{d_{k-1}\alpha}
$$

where the second inequality follows from the assumption $\sup_{y\in\mathcal{Y}} |y| \leq 1$, and third and the forth inequalities are due to the triangle inequality and Cauchy-Schwartz, respectively. Plugging the above results in Theorem 4 and using union bound we get

$$
L(\mathrm{w}) \leq \widehat{L}(\mathrm{w}) + C(1 + \sqrt{\alpha d_{k-1}})\sqrt{\frac{\alpha d_2}{n}} + C'(1 + \sqrt{d_{k-1}\alpha})^2\sqrt{\frac{\log(2/\delta)}{2n}}
$$

holds with probability at least $1 - 2\delta$, which completes the proof. $\qquad\square$

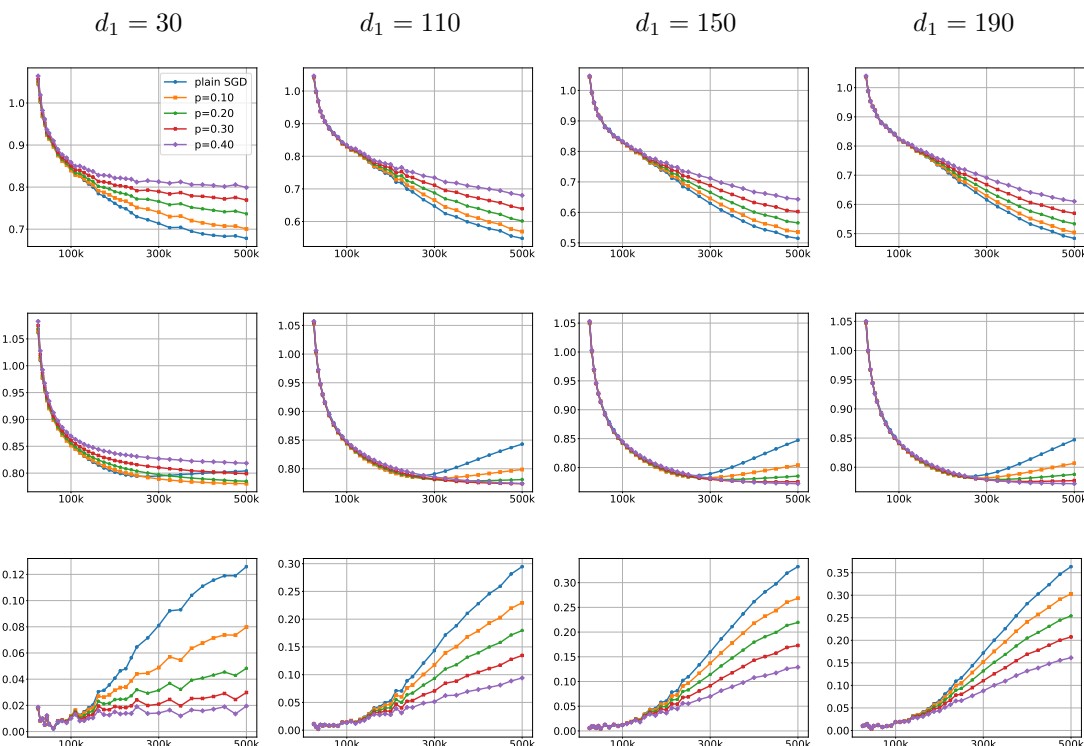

Figure 1: MovieLens dataset: the training error (**top**), the test error (**middle**), and the generalization gap for plain SGD as well as dropout with $p \in \{0.25, 0.50, 0.75\}$ as a function of the number of iterates, for different factorization sizes $d_1 = 30$ (first column), $d_1 = 110$ (second column), $d_1 = 150$ (third column), and $d_1 = 190$ (forth column).

## D    ADDITIONAL EXPERIMENTS

In this section, we include additional plots which was not reported in the main paper due to the space limitations. Figure 1 in the main paper shows comparisons between plain SGD and the dropout algorithm on the MovieLens dataset for a factorization size of $d_1 = 70$. The observation that we make with regard to those plots is not at all limited to the specific choice of the factorization size. In Figure 1 here, we report similar experiments with factorization sizes $d_1 \in \{30, 110, 150, 190\}$. It can be seen that the overall behaviour of plain SGD and dropout are very similar in all experiments. In particular, plain SGD always achieves the best training error but it has the largest generalization gap. Furthermore, increasing the dropout rate increases the training error but results in a tighter generalization gap.

It can be seen that an appropriate choice of the dropout rate *always* perform better than the plain SGD in terms of the test error. For instance, a dropout rate of $p = 0.2$ seems to always outperform plain SGD. Moreover, as the factorization size increases, the function class becomes more complex, and a larger value of the dropout rate is more helpful. For example, when $d_1 = 30$, the dropout with rates $p = 0.3, 0.4$ fail to achieve a good test performance, where as for larger factorization sizes ($d_1 \in \{110, 150, 190\}$), they consistently outperform plain SGD as well as other dropout rates.

