# OpenReview forum: "Dropout: Explicit Forms and Capacity Control"
_ICLR.cc/2020/Conference — Reject_

### Official Review · AnonReviewer2 · 2019-10-10
**Official Blind Review #2**

**Rating:** 1

**Review:**

The authors prove bounds on the generalization of models produced using dropout.  They conduct experiments showing that dropout improves over SGD without dropout, and plotting generalization gaps and their bounds.

Much of the technical leverage exploited in this paper comes from earlier work.  Their acknowledgement of these contributions is somewhat uneven.  For example, results a lot like Proposition 2 can be found in [1].  In their treatment
of deep networks, because only the last layer is dropped out, dropout is essentially applied to a linear model,
and Proposition 3 of this paper  follows from (10) of [2], which was pointed out 12 lines below (10) in that paper.

The statement of Theorem 1 does not appear to be
rigorous to me.  For random data, the probability
that any minimizer of the dropout ERM objective
satisfies their bounds on the lengths of the
rows of U and V could be less than 1 - 2 delta, in which case in some
cases where the displayed equation in Theorem 1
is said to apply, there is no U and V to apply it to.
(The parameter gamma is not quantified in the statement
of that theorem.  It is conceivable to me that if gamma is
constrained to be large, possibly relative to d_0, d_1 and d_2,
then the statement of the theorem could make sense.)

Theorem 2 has a similar issue.  How is M quantified?
How do we know that a minimizer that satisfies the constraints
on M exists with probability at least 1 - 2 delta?

The authors' claim that "changing the learning rate or the batch size does not significantly improve the
performance of any of these algorithms" is a little hard to believe.  My impression is that these choices
affect the implicit regularization of SGD (along with the initialization).  Some more detail about what
they tried would be helpful.

There is some interesting new content in the paper, even if, on the whole, it is a bit conceptually and technically
incremental.

(This review has been edited in light of the response.)


[1] Cavazza, Jacopo, et al. "Dropout as a Low-Rank Regularizer for Matrix Factorization." International Conference on Artificial Intelligence and Statistics. 2018.

[2] Wager, Stefan, Sida Wang, and Percy S. Liang. "Dropout training as adaptive regularization." Advances in neural information processing systems. 2013.


**Experience Assessment:**

I have published one or two papers in this area.

**Review Assessment: Checking Correctness Of Derivations And Theory:**

I assessed the sensibility of the derivations and theory.

**Review Assessment: Checking Correctness Of Experiments:**

I assessed the sensibility of the experiments.

**Review Assessment: Thoroughness In Paper Reading:**

I read the paper thoroughly.

---

> ### Author Response · Authors · 2019-11-15
> **Author response (see general comments above)**
>
> Regarding “Theorem 1 does not appear to be rigorous ”, and “the probability
> that any minimizer of the dropout ERM objective satisfies their bounds on the lengths of the rows of U and V could be less than 1 - 2 delta”.
> Gamma is chosen post hoc, after the training finished, as is conventional in data-dependent bounds.
>
> “How is M quantified?” As clearly stated in the theorem, M is an upper bound on the product of the norms.
>
> “When they plot their bounds in the experimental section, what value of gamma do they use?” → we do not plot the theoretical bound in those plots at all.
>
> Regarding “Proposition 2 can be found in [1]”, not true, see our general comments above. We also note that there are problems with the claims of the paper by Cavazza et al. The authors state in the abstract that “we prove that dropout is equivalent to a convex approximation problem with (squared) nuclear norm regularization”, and they have a formal statement in Theorem 2 to prove that claim. However, there is no proof of it in the supplementary. Instead, they have another Theorem 1 in the supplementary which shows a sufficient condition for local minima  to be a global minima but their claim of the necessary condition makes no sense. We know from previous work of (Mianjy et al., 2018) that the only way to show Theorem 2 is to show that “equalization” of factors is necessary and sufficient for optimality. There is no such notion in the Cavazza paper, and we are very confident that they cannot prove Theorem 2. We do cite (Mianjy et al., 2018).
>
> “In their treatment of deep networks, because only the last layer is dropped out, dropout is essentially applied to a linear model”. This is not true and grossly misleading. If this were the case, we could simply borrow known generalization bounds. There is a big difference between the complexity of a linear model that acts on the given input features, and a linear model on the features that are simultaneously trained. In fact, what we get in linear model is data dependent ell_2 norm which is useless as it amounts to scaling of parameters as we discuss in Section 3.1. Whereas for neural networks we get (data dependent) path norm which has been shown to provide size independent capacity control.
>
> Regarding proposition 3 itself, we agree that it is an elementary result, something we need along the way to get to the main result. We do not argue that it is a contribution. We already list our main contributions in the paper. Yes, we can find it in Wager et al. which we cite.

---

### Official Review · AnonReviewer1 · 2019-10-24
**Official Blind Review #1**

**Rating:** 1

**Review:**

Post Discussion Update:

The authors vehemently disagree with my critiques about discussion of / attribution to prior work.  They seem to think that the differences from Cavazza et al. [AIStats 2018] would be obvious to "even someone who has taken a basic course in machine learning."  However, both Reviewer #2 and Reviewer #3 mentioned the same issues, saying "much of the technical leverage exploited in this paper comes from earlier work...results a lot like Proposition 2 can be found in [Cavazza et al.]" (#2), and "I am surprised that [Cavazza et al.] is not cited with its due credit in the writing. I hope to see some concrete statement about the difference between the authors' contribution and the existing literature" (#3).  Since 3 out of 4 reviewers did not recognize a clear distinction, it is fair to say the differences to which the authors refer in their rebuttal should be discussed at length in the paper.  However, the authors have not posted a revised draft or any sample text , leaving me to keep my recommendation at 'reject.'

______________________________________________________________________________________________________________________________________
Summary:  This paper studies (Bernoulli) Dropout regularization in the context of matrix sensing and neural networks (when dropout is applied to only the last hidden layer).  The paper first focuses on matrix sensing, showing an explicit regularizer with connections to trace norm regularization and proving a generalization bound.  The paper then moves to neural networks, first showing an explicit regularizer in the case of a squared loss and dropout on the last hidden layer only.  When the input distribution is symmetric and isotropic, the explicit regularizer is shown to have connections to path norm regularization.  From this explicit regularizer, the authors then derive an upper bound on the generalization gap (Theorem 2).  Experiments are reported that show (#1) (stochastic) dropout does improve generalization in a matrix completion task and (#2) the theoretical results do predict generalization as tested on MNIST and CIFAR-10.

Pros:  Extending dropout to other tasks and understanding its general method of action is an important problem, thus making the paper well motivated.  Moreover, the approach of deriving explicit regularizers from the stochastic objective is a commendable strategy that could improve the stability and speed of converge.  Furthermore, understanding the generalization properties of neural networks is important, and as dropout seems to be a well-established tool for improving generalization, the paper’s approach is sensible.

Cons:  I find this paper to severely over-claim its contributions---in particular #1 and #3 from the introduction...

(#1) Dropout for matrix completion:  Claimed contribution #1 is incremental as it merely adapts the dropout strategy of Cavazza et al. [AIStats 2018] to matrix sensing.  Cavazza et al. [AIStats 2018]’s procedure “drop[s] columns of the factors” (a quote from their abstract), and this is exactly what is done in this paper: “...a procedure that randomly drops the columns of the factors during training” (p 1).  I find it suspicious that the only time Cavazza et al. [AIStats 2018]’s work is mentioned is in the Introduction’s long list of citations of previous dropout work.  In effect, this equates Cavazza et al. [AIStats 2018]’s work with much less related work (e.g. Bayesian interpretations).  The Cavazza et al. [AIStats 2018] work should surely be cited in the vicinity of Equation 2.  Furthermore, Cavazza et al. also discuss connections to trace norm regularization (Section 4) and should also be included in the paper’s discussion on page 3.

(#3) Dropout in NNs: The explicit regularizer derived in Prop 3 was previously derived by Wang & Manning [ICML 2013]; see their Section 3.1.  This work is not cited---another significant oversight.  However, the resulting complexity bounds derived from Wang & Manning [ICML 2013]’s explicit regularizer are original, to the best of my knowledge.

In general, the paper makes several claims that are at best ungenerous to previous work.  For instance, the Introduction claims “none of these [previous] works adequately address the following basic question: how does dropout control the capacity of deep neural networks?” (p 1).  This is an unsubstantiated claim given that the paper contains no Related Work section, which is needed since there have been dozens of papers written on dropout.  Moreover, this work considers only the case in which dropout is applied to the last hidden layer, not to the full network (which is a valid and sensible restriction).  Yet this essentially reduces the results to a study of dropout in linear models (except perhaps in Prop 4) and therefore I don’t see how one could claim the previous work of Wager et al. [NeurIPS 2013], which also studies dropout for linear models, doesn't address similar questions.  For another example, the paper claims on page 3: “These observations are specifically important because they connect dropout, an algorithmic heuristic in deep learning, to strong complexity measures that are empirically effective as well as theoretically well understood.”  I agree that the connections are important, but similar connections have already been established by (at least) Cavazza et al. [AIStats 2018], Wang & Manning [ICML 2013], and Wager et al. [NeurIPS 2013].  Such connections are not unique to this paper, as the text implies.

As for experiments, the matrix completion results to not validate any of the results in Section 2.  A comparison of training under the stochastic objective vs with the explicit regularizer is never performed (as is done in Cavazza et al. [AIStats 2018]).  Similarly, the generalization bounds are not shown to be useful.  The only thing that is shown is that the stochastic objective (again, which is a minor adaptation from Cavazza et al. [AIStats 2018]) does improve generalization.

Minor comments:

> ERM is never defined as an initialism for “empirical risk minimization”

> While the assumptions of an isotropic and symmetric input distribution in Prop 4 are unrealistic in general, such conditions would be satisfied by hybrid architectures defined by making the early layers of the network a (isotropic Gaussian) normalizing flow [Nalisnick et al., ICML 2019].

Final Evaluation:  I find the paper's only original and validated contribution to be using Wang & Manning [ICML 2013]’s explicit regularizer to derive the complexity upper bound in Lemma 1.  Due to the paper's lack of discussion and, at times, mischaracterization of previous work, the text needs to be significantly revised before it can be accepted.  A proper Related Work section must be added to discussion the previous literature on understanding dropout.



__References__

Nalisnick, Eric, et al. "Hybrid Models with Deep and Invertible Features." International Conference on Machine Learning. 2019.

Wang, Sida, and Christopher Manning. "Fast dropout training." International Conference on Machine Learning. 2013.


**Experience Assessment:**

I have published one or two papers in this area.

**Review Assessment: Checking Correctness Of Derivations And Theory:**

I assessed the sensibility of the derivations and theory.

**Review Assessment: Checking Correctness Of Experiments:**

I carefully checked the experiments.

**Review Assessment: Thoroughness In Paper Reading:**

I read the paper thoroughly.

---

### Official Review · AnonReviewer4 · 2019-10-31
**Official Blind Review #4**

**Rating:** 1

**Review:**

Summary:
This work studies the effect of regularization through dropout on generalization bounds for Matrix Sensing and linear regression with deep neural network task. For both matrix sensing task and linear regression task, authors derive an explicit regularization term due to dropout. Authors give elegant interpretations of dropout regularizer. For matrix sensing: dropout can be thought of as the inducing trace-norm regularization when matrices are standard gaussian. For matrix completion: dropout can be thought of as weighted trace-norm regularizer.
Authors then give a generalization bound for matrix completion with dropout.
In the context of deep neural networks, under assumptions on the input distribution authors show that the explicit regularizer associated with dropout is exactly the squared l2 path-norm of the network.
Authors give a bound on the Rademacher complexity of deep nerual networks with dropout. Using the bound on Rademacher complexity, authors derive the a generalization bound for squared error loss.

I recommend rejecting this submission. Following are my concerns:

1. Authors derive the bound on the Rademacher complexity in Lemma 1 and claim that this bound holds for any neural network, but in the proof they make assumptions on the output of the neural network that are not stated clearly. Assumptions, as far as I understand, are 1) expected output under the data distribution of the jth unit is bounded by 1 and 2) the second moment is also equal to 1. I believe that this is not true for most of the neural networks with RELU nonlinearity for the output units.

2. For the proof of Theorem 2, authors derive the supremum deviation between the true labels and those predicted by the neural network. I don’t believe the 5th inequality on page 19. Authors bound the worst case output of the neural network by the expected output of the neural network which is not true.

3. I am not yet convinced by the experiments section of the paper where they evaluate the generalization gap for datasets. In particular, the theorems derived in the paper has assumptions on the sample complexity. These assumptions are not verified for the datasets used in the experiment section and I suspect that lower bound on n can be too large for the bounds to be meaningful.

4. There is no comparison to existing literature on generalization bounds due to dropout as a regularizer. Following paper derives PAC-Bayes bound for dropout procedure:
	McAllester, D.A. (2013). A PAC-Bayesian Tutorial with A Dropout Bound. ArXiv, abs/1307.2118.

**Experience Assessment:**

I have read many papers in this area.

**Review Assessment: Checking Correctness Of Derivations And Theory:**

I carefully checked the derivations and theory.

**Review Assessment: Checking Correctness Of Experiments:**

I assessed the sensibility of the experiments.

**Review Assessment: Thoroughness In Paper Reading:**

I read the paper at least twice and used my best judgement in assessing the paper.

---

> ### Author Response · Authors · 2019-11-15
> **Point by point response (also see the general comments above)**
>
> 1. We do not make any implicit assumption as suggested. In the proof of Lemma 1 on page 17, if the reviewer would look carefully, just before the first inequality in the beginning of the proof, we multiply and divide by a_j. So the function that we define as h_w is bounded in L2 norm. Again, this is by construction, using our analysis technique, we do not need this to hold for any function, but for the purposes of the proof we normalize the function implemented by a subnetwork and thus the property we need holds. All the assumptions about boundedness of data (which are necessary) are clearly stated in the statement of Theorem 2.
>
> 2. Yes, that is correct. We did find that issue soon after submitting and there is an easy fix. As is common in the learning theory, we can apply a 1-Lipshitz clipping function that thresholds the output function in [-1, +1] which is anyway the range of the label space. In fact, this change improves the bound.
>
> 3. That is right; the lowerbound on n might be too large in practice. We do not claim that these bounds are tight; however, we do observe a nice correlation between the bounds and the actual gap.
>
> 4. We are not aware of generalization bounds for dropout in matrix sensing and in ReLU networks. Also see the general comment above regarding comparison with other works.

---

### Official Review · AnonReviewer3 · 2019-11-10
**Official Blind Review #3**

**Rating:** 3

**Review:**

The paper is well written, but severely overestimates the core contributions embedded in section 3.

Firstly, the idea of using drop out for matrix sensing seems to be a somewhat trivial extension of the work on dropout for matrix factorization -- http://proceedings.mlr.press/v84/cavazza18a/cavazza18a.pdf.  I am surprised that this work is not cited with its due credit in the writing. I hope to see some concrete statement about the difference between the authors' contribution and the existing literature. It is fine to propose an incremental improvement as long as the original work receives the required credit.

Dropout has been an extensive area of theoretical research for the past few years. The overly simplistic statement about the limitation of our understanding of how dropout works are a little disappointing, particularly so when the authors themselves list this literature in the related work section. That dropout training can be perceived as an adaptive regularization is also very well known and researched. These arguments weaken the second and third contribution of the paper. I do understand though that extending the results of deep linear networks to a single hidden layer RELU network is non-trivial. The derivations also suggest so and the authors deserve credit for attempting these derivations.

I am also quite doubtful about the setting of the experiments in section 4.1. That changing the batch-size or learning rate does not significantly influence the eventual performance is very counter-intuitive.

Overall, the paper appeared quite promising in the beginning, but the claims in the introduction are not well supported through the rest of the paper.


**Experience Assessment:**

I have read many papers in this area.

**Review Assessment: Checking Correctness Of Derivations And Theory:**

I assessed the sensibility of the derivations and theory.

**Review Assessment: Checking Correctness Of Experiments:**

I assessed the sensibility of the experiments.

**Review Assessment: Thoroughness In Paper Reading:**

I read the paper at least twice and used my best judgement in assessing the paper.

---

### Author Response · Authors · 2019-11-15
**Author response**

Comparison with Cavazza et al.: We disagree that we did not discuss the previous work of Cavazza et al. appropriately. Matrix factorization is not a learning problem. What are the inputs, what are the outputs, what is the distribution over in the matrix factorization problem; is there even anything stochastic in matrix factorization?  We argue that matrix factorization is a numerical optimization problem in linear algebra, vastly different from any learning problem we consider. Matrix sensing and matrix completion, on the other hand, are among the most important matrix learning problems, both of which we study in this paper. Yes, the nature of results we show are similar, that the induced regularizer due to dropout is nuclear norm. However, this similarity is very superficial. What is the role of a regularizer in a numerical optimization problem? Besides, Cavazza et al. only show a derivation of explicit regularizer (which by the way is elementary). They do not show that at the minimizers of the matrix factorization problem the regularizer acts like nuclear norm, they only show that the convex envelope of the explicit regularizer is nuclear norm. In other words, the very problem setup, the nature of statements of paper in Cavazza et al. and the tools we use are different. Our proofs are simple and quickly verified. We do build on prior work of (Mianjy et al., 2018) which studied the regression problem with single hidden layer linear networks and we do credit (Mianjy et al., 2018) adequately, and even produce Theorem statements from that paper verbatim with proper citation.

In summary, we rigorously argue for dropout in matrix completion by 1) showing the induced regularizer is equal to weighted trace-norm (novel result) 2) give a generalization bound 3) provide extensive experimental evidence that dropout provides state of the art performance on one of the largest datasets in recommender systems research. What is an example of dropout being useful in matrix factorization?

Novelty/Significance: The novelty is in understanding dropout for learning problems. None of the prior work says anything about dropout for matrix completion. Beyond that we rigorously extend our results to neural networks, give explicit regularizer, bound Rademacher complexity of norm bounded hypotheses, show precise generalization bounds, and support them with empirical results. We are not aware of generalization bounds for dropout in matrix sensing and in ReLU networks. Reviewer 1 says that we “merely adapt the dropout strategy of Cavazza et al. ”, “The Cavazza et al. work should surely be cited in the vicinity of Equation 2”. Dropout strategy is THE dropout strategy in the original dropout work of Srivastava. We are not claiming any credit for proposing Dropout. Even for matrix factorization, it has been long studied before Cavazza et al. (e.g. here: https://arxiv.org/pdf/1512.04483.pdf)

Incremental work? How is giving generalization bounds for learning problems (including regression with deep neural networks) an incremental improvement over Cavazza et al. which does not even focus on any learning problem, only focuses on matrix factorization, considers no non-linearity in their models? When does not citing a paper in “vicinity” of a certain equation/result become a basis for rejecting a paper? Even if that were a reasonable request (it is not, see above) is it really something that warrants a rating of 1 (strong reject)?

Comparison with other work: We do cite most of the previous work on Dropout. There are hardly any instances of papers showing any kind of explicit complexity/capacity control and that is what we meant. Adding a detailed discussion of every work on Dropout when the focus of our investigation is so different does not seem like a useful exercise.

Regarding the two specific papers suggested by the reviewers, the paper by Wang and Manning only considers linear predictors, their result for linear regression can be found even in the original dropout paper (see section 9.1. http://jmlr.org/papers/volume15/srivastava14a.old/srivastava14a.pdf). The result by David McAllester is no different as it shows that Dropout in linear regression amounts to \ell_2 regularization. Nature of regularization in linear predictors versus predictors in factored form (linear or nonlinear) are quite different as we discuss in Section 3.1.

Overall: There are very few comments regarding the technical content and contributions of the paper. The harsh ratings are not justified by arguments made by the reviewers. It is unnecessary to say that we are saddened by the current state of the affair, but it is clear what we need to do for the next version of the paper. We will discuss the differences above in great detail even if it is unnecessary, to make sure we cover all our bases. Thank you!

---

> ### Comment · AnonReviewer1 · 2019-11-15
> **Re Author Response**
>
> Thank you for your response, authors.
>
> You say that "We disagree that we did not discuss the previous work of Cavazza et al. appropriately," but the problem is that you did not discuss them *at all.*  You admit yourselves that "Yes, the nature of results we show are similar," so why not explicitly discuss the differences in a Related Work section?  The distinctions you make in your rebuttal are too subtle to leave to the reader.
>
> "Dropout strategy is THE dropout strategy in the original dropout work of Srivastava. We are not claiming any credit for proposing Dropout":  In contribution #1 in the Introduction, you state: "We introduce dropout for matrix completion..."  This reads like a claim to novelty, whether you intend it to or not.
>
> While I regret that the authors feel like we did not discuss "the technical content and contributions of the paper," from my perspective I was discussing the "contributions" w.r.t. to their novelty and originality.  Maybe there is a distinction to be had in your discussion of "learning problems" vs "numerical optimization," but the current draft does not make that clear.

---

> > ### Author Response · Authors · 2019-11-15
> > **Learning vs optimization**
> >
> > We cite Cavazza et al. for previous work related to understanding Dropout. There is not much else to say because we are interested in statistical learning. Hence it is appropriate to cite it as we did. This is what we meant. You argue that we do not cite it "appropriately", we disagree because all that was needed was to mention it.
> >
> > You copied a phrase out of our paragraph to suit your narrative. The purpose of the rest of the paragraph if you go back and read it is that the similarity you talk about is only superficial, again, which is why there is no need to refer to it anymore besides referring to it as we did. The work that is relevant is that of regression with single hidden layer linear networks, where the induced regularizer is also the Nuclear norm. We give credit where it is due.
> >
> > We do not have "our" definition of learning. Even someone who has taken a basic course in machine learning knows that it is about generalization. What is the notion of generalization in matrix factorization? What is the test data for the matrix factorization problem?
> >
> > What is regretful is that the supposed "subject matter expert" does not know the distinction.

---

### Decision · Program_Chairs · 2019-12-19

**Decision:**

Reject

**Comment:**

The authors study dropout for matrix sensing and deep learning, and show that dropout induces a data-dependent regularizer in both cases. In both cases, dropout controls quantities that yield generalization bounds.

Reviewers raised several concerns, and several of these were vehemently rebutted. The rhetoric of the back and forth slid into unfortunate territory, in my opinion, and I'd prefer not to see this sort of thing happen. On the one hand, I can sympathize with the reviewers trying to argue that (un)related work is not related work. On the other hand, it's best to be generous, or you run into this sort of mess.

In the end, even the expert reviewers were unswayed. I suspect the next version of this paper may land more smoothly.

While many of the technical issues are rebutted, one that caught my attention pertained to the empirical work. Reviewer #4 noticed that the empirical evaluations do not meet the sample complexity requirements for the bounds to be valid (nevermind loose). The response suggests this is simply a fact of making the bounds looser, but I suspect it may also change their form in this regime, potentially erasing the empirical findings. I suggest the authors carefully consider whether all assumptions are met, and relay this more carefully to readers.